# Engineering of the CHAPk Staphylococcal Phage Endolysin to Enhance Antibacterial Activity against Stationary-Phase Cells

**DOI:** 10.3390/antibiotics10060722

**Published:** 2021-06-16

**Authors:** Sara Arroyo-Moreno, Máire Begley, Kornelia Dembicka, Aidan Coffey

**Affiliations:** 1Department of Biological Sciences, Munster Technological University, Cork T12 P928, Ireland; sara.arroyo-moreno@mycit.ie (S.A.-M.); maire.begley@cit.ie (M.B.); kornelia.dembicka@mycit.ie (K.D.); 2APC Microbiome Institute, University College, Cork T12 YT20, Ireland

**Keywords:** endolysins, *S. aureus*, LysK, lysostaphin, biofilms, phage therapy

## Abstract

Bacteriophage endolysins and their derivatives have strong potential as antibacterial agents considering the increasing prevalence of antibiotic resistance in common bacterial pathogens. The peptidoglycan degrading peptidase CHAPk, a truncated derivate of staphylococcal phage K endolysin (LysK), has proven efficacy in preventing and disrupting staphylococcal biofilms. Nevertheless, the concentration of CHAPk required to eliminate populations of stationary-phase cells was previously found to be four-fold higher than that for log-phase cells. Moreover, CHAPk-mediated lysis of stationary-phase cells was observed to be slower than for log-phase cultures. In the present study, we report the fusion of a 165 amino acid fragment containing CHAPk with a 136 amino acid fragment containing the cell-binding domain of the bacteriocin lysostaphin to create a chimeric enzyme designated CHAPk-SH3blys in the vector pET28a. The chimeric protein was employed in concentrations as low as 5 μg/mL, producing a reduction in turbidity in 7-day-old cultures, whereas the original CHAPk required at least 20 μg/mL to achieve this. Where 7-day old liquid cultures were used, the chimeric enzyme exhibited a 16-fold lower MIC than CHAPk. In terms of biofilm prevention, a concentration of 1 μg/mL of the chimeric enzyme was sufficient, whereas for CHAPk, 125 μg/mL was needed. Moreover, the chimeric enzyme exhibited total biofilm disruption when 5 μg/mL was employed in 4-h assays, whereas CHAPk could only partially disrupt the biofilms at this concentration. This study demonstrates that the cell-binding domain from lysostaphin can make the phage endolysin CHAPk more effective against sessile staphylococcal cells.

## 1. Introduction

*Staphylococcus aureus* is a common member of the human skin flora, being also an opportunistic pathogen in clinical and healthcare settings [1]. The history of *S. aureus* treatment has been marked by the development of resistance to several antibiotics. Many clinical isolates are resistant to methicillin and almost all β-lactam antimicrobials [2]. These antibiotic-resistant forms are classified as “high priority” pathogens by the World Health Organization (WHO, Geneva, Switzerland) [3]. These bacteria can cause a wide variety of diseases commonly involving the skin and soft tissue. It can also be associated with other clinical infectious syndromes, in which bacteria can be dividing very slowly or not dividing at all, including bacteremia, endocarditis, intravascular infections, pneumonia, osteomyelitis, septic arthritis, pyomyositis, necrotizing fasciitis, orbital infections, endophthalmitis, parotitis, staphylococcal toxinoses, urogenital infections, and central nervous system infections [2]. Many of these syndromes involve biofilm-associated infections, in which cells are in the stationary phase, presenting a higher tolerance to different antibiotics [4]. Staphylococcal cells are subject to physiological changes when they enter the stationary phase, including an increase in the peptidoglycan thickness and in the periplasmic space and reduction in membrane fluidity [5]. These physiological conditions involve interferences with transcription, translation, or ATP synthesis and dramatically increase cell survival and persistence [6,7,8]. Cell surface-associated proteins are also expected to play a role when cells switch into a persister state, as they form the first line of molecular interaction with substances in the environment [9]. 

Bacteriophages (phages) and their derived peptidoglycan lytic enzymes are promising alternatives to conventional antibiotics for the inhibition of *S. aureus* [10]. Endolysins or lysins are phage-encoded peptidoglycan hydrolases employed by most phages to enzymatically degrade the peptidoglycan layer of the host bacterium at the end of their lytic multiplication cycle to the release the viral progeny [11]. Different studies have previously described the efficacy of endolysins to eliminate bacterial cells in the biofilm state [12,13,14,15,16,17]. However, other studies have shown that stationary phase cells are less susceptible to endolysins due to peptidoglycan maturation [18,19,20]. CHAPk is a truncated derivate of the staphylococcal endolysin LysK, encoded by the phage K [21,22]. Lysostaphin is a bacteriocin produced by *S. simulans* with antimicrobial activity against *S. aureus*. Previous studies proved the efficacy of CHAPk against log-phase planktonic cultures and biofilms [23,24], and the synergy between CHAPk and lysostaphin [25,26,27]. Phage endolysins are known to be highly refractory to resistance development, and no bacterial resistance has been reported [28]. Nevertheless, resistance to lysostaphin has been reported [29,30]. Previous studies demonstrated how the SH3b cell-binding domain of lysostaphin could improve the lytic activity of staphylococcal lysins [12,27,31,32,33,34]. The SH3b cell-binding domain of lysostaphin possesses an unusual binding mechanism that allows a synergistic and structurally dynamic recognition of *S. aureus* peptidoglycan, as the pentaglycine cross-bridge and the peptide stem are recognized by two independent binding sites located on opposite sides of the SH3b domain [35]. For example, the study performed by Becker et al. in 2009 [31] showed that the cell-binding domain from lysostaphin could increase more the anti-staphylococcal activity of the endolysin λSa2 compared to the cell-binding domain from LysK, presenting statistically significant turbidity reduction rates. 

In this study, apart from demonstrating again how the cell-binding domain of lysostaphin can improve the lytic activity of an endolysin, we proved that the unique mechanism of this cell-binding domain [35,36] could also make a phage endolysin more efficient against non-dividing cultures and biofilms.

## 2. Results

### 2.1. Comparison of Activity of CHAPk with Lysostaphin

A synergistic effect could be observed when combining different concentrations of CHAPk and lysostaphin using 7-day-old cultures (Figure 1). In addition, lysostaphin was found to be as active on 7-day old cultures as on log-phase cells, while CHAPk was not (data not shown). One key difference between the two enzymes was the presence of the cell-binding domain (SH3blys) on lysostaphin and it was of interest to confirm if this was responsible for the activity on the stationary-phase cells. 

### 2.2. Production and In Vitro Activity of CHAPk and CHAPk-SH3blys

The chimeric protein based on both enzymes, namely CHAPk-SH3blys, was generated with the fusion of CHAPk and SH3blys from lysostaphin, using the *E. coli* vector pET28a, as shown in Figure 2. 

Recombinant lytic enzymes were purified by affinity chromatography to >90% homogeneity. The yield obtained was approximately 5 mg of purified protein per liter of culture. When purified protein fractions of CHAPk were examined by SDS–PAGE, a band of approximately 20 kDa was evident, which corresponds to the predicted molecular mass of 18.6 kDa (Figure 3). In the case of CHAPk-SH3blys, a band of approximately 35 kDa was observed, which corresponds to the predicted molecular mass of 33.6 kDa (Figure 3). CHAPk and CHAPk-SH3blys fractions were also examined for lytic activity by zymogram analysis where zones of cell lysis were evident at the 20 kDa and 35 kDa position, respectively (Appendix A).

### 2.3. Turbidity Reduction Assays

Different concentrations of CHAPk were used against *S. aureus* DPC5246 log-phase and 7-day-old cells both resuspended in fresh endolysin buffer. A concentration of 5 μg/μL (268 nM) of CHAPk could reduce the OD_590_ from 0.3 to 0.2 in 20 min in the log-phase cell suspension, but not in the suspension containing the 7-day-old cells (Figure 4). Treatment of the 7-day-old cells with higher concentrations of CHAPk did result in a decrease in OD_590_ (Figure 4). Different concentrations of CHAPk-SH3blys (5, 20, and 50 μg/mL; i.e., 148.8 nM, 595.2 nM, and 1.5 μM respectively) were used as well against log-phase and 7-day-old cultures (Figure 4). A concentration of 5 μg/μL (148.8 nM) of CHAPk-SH3blys can reduce approximately 0.1 of the OD_590_ in a 7-day-old culture of *S. aureus* DPC5246 in 20 min (Figure 4). 

### 2.4. Colony Plate Count after Turbidity Reduction Assays

After performing turbidity reduction assays on log-phase and 7-day-old cells with different concentrations of CHAPk and CHAPk-SH3blys, a viable plate count assay was performed to confirm that the reduction in turbidity was related to loss of cell viability (Figure 5). 

There was a 2-log decline when the 7-day-old cultures were treated with 20 and 50 μg/mL of CHAPk, resulting in a viable plate count of 3.41 × 10^5^ CFU/mL and 4.96 × 10^5^ CFU/mL, respectively (from a starting titer of 2.25 × 10^8^ CFU/mL). Reduction in CFU/mL was not observed when the 7-day-old cultures were treated with 5 μg/mL of CHAPk. In contrast, a 4-log decrease was seen for log-phase cells when treated with 50 μg/mL of CHAPk, a 3-log reduction was observed when the concentration of CHAPk was 20 μg/mL, and a 2-log reduction resulted from treatment with 5 μg/mL of CHAPk (Figure 5). This demonstrated that the log-phase cells were more susceptible to the lytic activity of CHAPk. 

*S. aureus* DPC5246 7-day-old cultures that were treated with 20 and 50 μg/mL of CHAPk-SH3blys resulted in a 2-log reduction in viable plate count from a starting titer of 1.7 × 10^7^ CFU/mL to 7.8 × 10^5^ CFU/mL and 1.3 × 10^5^ CFU/mL, respectively. A 1-log reduction was observed when the 7-day-old cultures were treated with 5 μg/mL of CHAPk-SH3blys, resulting in a viable plate count of 3.4 × 10^6^ CFU/mL. In contrast, a 4-log decrease was seen for log-phase cells when treated with 50 μg/mL of CHAPk-SH3blys, resulting in a viable plate count of 2 × 10^4^ CFU/mL. A 3-log reduction was observed when the concentrations of CHAPk-SH3blys were 5 and 20 μg/mL resulting in a viable plate count of 3.4 × 10^5^ CFU/mL and 1.2 × 10^5^ CFU/mL, respectively. All these results are summarized in Figure 5.

### 2.5. Minimal Inhibitory Concentration (MIC) Assays 

MIC assays were performed using an initial inoculum of 1 × 10^5^ CFU/mL log-phase or 7-day-old *S. aureus* DPC5246 cells. In the case of CHAPk, the MIC was found to be 4-fold lower when log-phase cultures were used as initial inoculum, compared to an initial inoculum of 7-day-old cultures (Table 1). In the case of CHAPk-SH3blys, the MIC for both log-phase and 7-day-old cultures was the same, being 4-fold lower compared to CHAPk against log-phase cultures and 16-fold lower compared to CHAPk against 7-day-old cultures (Table 1). Moreover, aliquots of 100 μL from clear wells were plated on mannitol salt agar (MSA), and no growth was observed after overnight incubation at 37 °C. 

### 2.6. Staphylococcal Biofilm Cultivation

The ability of *S. aureus* DPC5246 to form biofilm on plastic was examined using a standard microtiter-based assay where any biofilm that was formed was quantified by staining with crystal violet and measuring optical density. Biofilms were formed when both log-phase cells and 7-day-old cells were used as initial inoculum. It was noted that readings were higher for the 7-day-old cells (Appendix A). 

### 2.7. Staphylococcal Biofilm Prevention Using CHAPk and CHAPk-SH3blys

CHAPk was assessed for its effectiveness in preventing biofilm formation by log-phase and 7-day-old cultures of *S. aureus* DPC5246 cells in triplicate (Figure 6). This was determined by adding varying concentrations of CHAPk ranging from 0.5 to 125 μg/mL and comparing the result obtained to the control, i.e., media inoculated with bacteria to which CHAPk was not added. A significant reduction in biofilm quantity was observed for all CHAPk concentrations for both log-phase and 7-day-old cells except for 0.5 μg/mL when the initial inocula were log-phase cultures. 

CHAPk-SH3blys was also assessed for its effectiveness in preventing biofilm formation by log-phase and 7-day-old cultures of *S. aureus* DPC5246 cells in triplicate. Similar to the results obtained for the experiments performed with CHAPk, a significant reduction in biofilm quantity compared to the control was also observed when log-phase and 7-day-old bacteria were incubated with all CHAPk-SH3blys concentrations except for 0.5 μg/mL when the initial inocula were log-phase cultures (Appendix A). CHAPk-SH3blys mediated significantly greater biofilm prevention compared to CHAPk, for both initial inocula, at concentrations ranging between 0.98 and 31.25 μg/mL (Figure 6). Results for biofilm prevention can also be observed visually (Appendix A). 

### 2.8. Staphylococcal Biofilm Disruption Using CHAPk and CHAPk-SH3blys

Having shown that the enzymes could prevent biofilm formation at certain concentrations, it was subsequently decided to examine if they could remove pre-established biofilm. We have previously shown that higher concentrations of CHAPk are required to remove established biofilm compared to the concentrations required for biofilm prevention [24]. Consequently, for the biofilm disruption experiments, higher enzymes concentrations were used than those used for the experiments outlined in Section 2.6. Biofilms were established using log-phase and 7-day-old *S. aureus* DPC5246 cells and then treated with concentrations of CHAPk ranging from 5 to 200 μg/mL (268.8 nM to 10 μM) for four hours in triplicate (Figure 7). CHAPk was most effective in disrupting biofilms formed from 7-day-old cultures cells when concentrations of 50 and 200 μg/mL were used, exhibiting a 14.8% and a 5.6% reduction in biofilm quantity, respectively. These reductions were statistically significant compared to untreated biofilm (*p*-values 0.0060 and 0.0430, respectively) (Appendix A). For the biofilms formed by log-phase cells, there was a statistically significant decrease in absorbance for all CHAPk concentrations tested (5 to 200 μg/mL) to (Figure 7). For biofilms formed from exponential phase cells, a 32.5% reduction in biofilm quantity was measured when the biofilms were treated with either 200 or 50 μg/mL of CHAPk and an 18.4% reduction in biofilm quantity was measured when the lowest concentration of CHAPk (5 μg/mL) was used for treatment. 

Biofilms cultivated using exponential and stationary phase *S. aureus* DPC5246 were also treated with CHAPk-SH3blys concentrations 5, 20, 50, and 200 μg/mL (148.8 nM, 595.2 nM, 1.5 μM, and 6 μM) for four hours in triplicate. CHAPk-SH3blys was most effective in disrupting biofilms formed by exponential and stationary phase cells when concentrations of 200 and 50 μg/mL were used, being the OD_595_ readings similar to the blank when exponential phase cells were used and exhibiting a 72% and 66% reduction in biofilm quantity, respectively, when biofilms were cultivated using stationary phase cells. Concentrations of 20 and 5 μg/mL could reduce approximately 40% of the biofilms formed by exponential phase cells, and they could reduce around 60% of the biofilms formed by stationary phase cells. All the concentrations of CHAPk-SH3blys could produce statistically significant biofilm disruption, for either biofilms formed from log-phase cultures or 7-day-old cultures (Appendix A). In Appendix A, it can also be observed how 7-day-old cultures produced a significantly higher biofilm biomass compared to log-phase cultures.

All concentrations of CHAPk-SH3blys produced a significantly higher biofilm disruption compared to the same concentrations of CHAPk when biofilms were induced from log-phase and 7-day-old cultures (Figure 7). Results for biofilm disruption can also be observed visually (Appendix A). 

## 3. Discussion

In the current study, the catalytic CHAP domain of the endolysin CHAPk was fused with the SH3b binding domain of lysostaphin, generating the chimeric enzyme CHAPk-SH3blys. This resultant enzyme was shown to be effective against stationary phase cells (7-day-old cultures), although log-phase cultures exhibited higher susceptibility. While others have shown how this binding domain increases activity against fresh cultures, we demonstrated the efficacy of this engineered protein against aged cultures, which is significant given that such cells are frequently the source of many chronic infections [37]. This study also demonstrates how the culture age-dependency activity observed in different endolysins can be overcome by protein engineering approaches. A combination of the lysostaphin SH3b cell-binding domain with different endolysin catalytic domains can be a way to improve staphylococcal antibiofilm activity and, therefore, generate enhanced antimicrobials to treat biofilm-related staphylococcal infections. CHAPk-SH3blys presented less culture-age-dependency in terms of lytic activity. This fact suggests that the lysostaphin SH3b cell-binding domain facilitates the catalytic domain to find its target bonds within the peptidoglycan, regardless of the presence of modifications in the matured peptidoglycan. For all the enzymatic concentrations tested for CHAPk and CHAPk-SH3b, in all cases, there was always a faster reduction in OD_600_ when CHAPk-SH3b was employed, compared to the same concentration of CHAPk. Moreover, lower concentrations of CHAPk-SH3blys could produce a reduction in turbidity against 7-day-old cultures compared to CHAPk. CHAPk-SH3blys is more inhibitory against *S. aureus* DPC5246 than CHAPk, regardless of the cell growth or metabolic state. Moreover, 7-day-old cultures produced stronger and more persistent biofilms, confirming findings that could already be seen in another study [38]. While both CHAPk and CHAPk-SH3blys were both shown to prevent and disrupt biofilms formed from either log-phase cultures or 7-day-old cultures, CHAPk-SH3blys was more effective than CHAPk. These results also confirm the findings from other studies, where combinations of different endolysin catalytic domains with the cell-binding domain from lysostaphin resulted in improved anti-staphylococcal activity. For example, in the study carried out by Paul et al. in 2011 [32], the cell-binding domain of lysostaphin was fused to the CHAP domain from the tail-associated hydrolase of phage K, and the activity achieved by this chimeric protein could only be matched by using a 100-fold higher concentration of a protein, consisting of only the CHAP domain. In another study, by Rodríguez-Rubio et al. [27], the same cell binding domain was combined with the CHAP domain from the virion associated hydrolase HydH5, resulting in higher staphylococcal activity than the parental enzyme by 64-fold. In our study, a 100-fold increase in activity was observed, in terms of biofilm prevention However, stationary-phase cells were not investigated in previous studies. 

In conclusion, our findings suggest that CHAPk-SH3blys may be a promising antimicrobial candidate to treat chronic infections caused by *S aureus*, as it can efficiently disrupt biofilms and lyse cells in late stationary phase. Moreover, the fact that it possesses a phage lysin catalytic domain makes it very unlikely that staphylococci will develop resistance to it. This chimeric enzyme is stable and active at physiological conditions, and it could be easily produced in large-scale. Some staphylococcal endolysins are currently at different phases of clinical trials [39,40,41]. Adverse effects have not been reported from any of these trials. Further studies should be performed on CHAP-SH3blys to assess its lytic activity in vivo and its safety to be used clinically.

## 4. Materials and Methods

### 4.1. Bacterial Strains and Growth Conditions

For this study, *S. aureus* DPC5246 (Teagasc, Cork) was used to assess enzymatic ability. This MRSA strain was routinely cultured in Brain Heart Infusion broth (BHI; Sigma Aldrich, Gillingham, UK) or Muller Hinton Broth (MHB; Sigma Aldrich) at 37 °C with shaking, as well as on BHI agar plates or mannitol salt agar (MSA) plates (Lab M, Heywood, UK). Biofilm cultivations of *S. aureus* DPC5246 were carried out using BHI broth supplemented with 1% glucose (BHIg). Recombinant *E. coli* BL21 (DE3) Star (Novagen, London, UK) containing the constructs pET28a-CHAPk and pET28a-CHAPk-SH3blys were routinely grown in Luria-Bertani (LB) broth, and agar (Sigma Aldrich) supplemented with 50 μg/mL of kanamycin (Sigma Aldrich).

### 4.2. Cloning, Expression, and Purification of Recombinant Phage-Derived Enzymes

The enzymes used for assessment of lytic activity were CHAPk (truncated derivate of the phage K endolysin lysK; Fenton et al., 2010), lysostaphin (from *Staphylococcus simulans*, Sigma), and CHAPk-SH3BLYS (combination of CHAPk and SH3 domain of lysostaphin). All these enzymes were suspended in 25 mM Tris-HCl pH 7 buffer.

The sequences encoding CHAPk and SH3blys were obtained from sequence accession numbers AY176327.1 and M15686, respectively. The chimera CHAPk-SH3blys and the CHAPk alone were both synthesized with *E. coli* codon optimization (GenScript, Leiden, The Netherlands and then inserted into the *Nco*I/*Xho*I site of the pET28a vector (Novagen) (Appendix A). Vectors containing inserts were transformed into BL21 (DE3) cells (Novagen), and then transformants were plated into LB agar plates containing 50 μg/mL of kanamycin. Both enzymes, CHAPk and CHAPk-SH3blys, were overexpressed and purified as follows. Transformant colonies containing either pET28a-CHAPk or pET28a-CHAPk-SH3blys were grown in test tubes containing LB broth supplemented with 50 μg/mL of kanamycin and incubated overnight at 37 °C with shaking. The following day, the overnight cultures were used to inoculate (1% inoculum) 1L conical flasks containing 500 mL of LB broth supplemented with 50 μg/mL of kanamycin. The conical flasks were incubated at 37 °C with shaking (160 rpm) until the OD_600_ of the cultures was 0.6–0.8, 0.5 mM IPTG (Sigma Aldrich) was added to the flasks, and they were incubated for 16 h at room temperature at 160 rpm. The whole contents of conical flasks were dispensed into several 100 mL centrifuge bottles, which were then centrifuged for 20 min at 4000× *g*. Bacterial pellets were resuspended in 2.5 mL of 50 mM sodium acetate, combined, and mixed with *Bug Busting* cells reagent, as per manufacturer instructions (Inovagen, Worcestershire UK). The resulting cell paste was transferred to a 50 mL falcon tube and incubated at room temperature at 20 rpm for 20 min. The cell pastes were centrifuged, and the supernatant containing the soluble fraction of the proteins was filtered sterilized (crude lysate). The crude lysates for CHAPk and CHAPk-SH3blys were subjected to His-tag affinity chromatography purification using His-Trap FF 1 mL columns for the AKTA *Start* system (GE healthcare, North Richland Hills, TX, USA). The purified proteins were concentrated and desalted by buffer exchange using an Amicon Ultra-15 Centrifugal Filter Unit with a molecular weight cut-off of 10 kDa (Sigma Aldrich)). The proteins were quantified using a BCA protein quantification kit (Fisher Scientific, Loughborough, UK), , as per manufacturer instructions) and then visualized in 12% SDS-PAGE gels and zymograms.

### 4.3. Turbidity Reduction Assays

A single colony of *S. aureus* DPC5246, from BHI agar plates, was incubated overnight at 37 °C with shaking in 10 mL of MHB. Then a 1% inoculum of the overnight cultures was transferred into fresh tubes containing 10 mL of MHB and then incubated at 37 °C at 160 rpm. In some tubes, cells were grown until they were log-phase (OD_600_ between 0.6–0.8), and other tubes were left in the shaking incubator for seven days. It was previously reported that staphylococcal cultures could remain in stationary phase for up to 7 days before entering death phase [42]. Log-phase and stationary phase (7-day-old cultures) planktonic cultures were centrifuged at 5000× *g* for 5 min, and cell pellets were resuspended with buffer 25 mM Tris-HCl pH 7 to an OD_600_ of 0.6–0.8. Turbidity reduction assays were carried out using a 96-well plate, in which 100 μL of cells were mixed with 100 μL of enzyme at different final concentrations (5, 20, and 50 μg/mL) or with 100 μL of 25 mM Tris-HCl pH 7 buffer, in triplicate. The reduction in absorbance at 590 nm was monitored using a plate reader (Spectra MAX 340, Molecular Devices, Molecular Devices, San Jose, CA, USA) at 37 °C for 20 min with OD_590_ readings taken every 5 min. 

### 4.4. Colony Plate Count after Turbidity Reduction Assays

Log-phase and 7-day-old cultures treated with final concentrations of 5, 20, or 50 μg/mL of CHAPk and CHAPk-SH3lys were subject to colony plate count after the 20 min turbidity reduction assays to confirm that the reduction in turbidity corresponds to the reduction of viable cells. Each well containing a specific concentration of CHAPk and CHAPk-SH3blys (50, 20 and 5 μg/mL), and wells treated only with 25 mM Tris-HCl buffer were serially 10-fold diluted. Diluted samples were plated onto MSA plates (Lab M) and plates were incubated at 37 °C overnight. 

### 4.5. Minimal Inhibitory Concentration (MIC) Assays

The MIC of CHAPk and CHAPk-SH3blys against *S. aureus* DPC5246 was determined according to the standard broth microdilution method of the Clinical and Laboratory Standards Institute [43]. Briefly, log-phase and 7-day-old cultures of strain DPC5246 (prepared as described in Section 2.5) were diluted in MHB to an initial inoculum of approximately 10^5^ CFU/mL. Serial 1:2 dilutions of CHAPk and CHAPk-SH3blys were performed in sterile flat-bottomed 96-well plates (Sarstedt) in MHB. Concentrations of CHAPk and SH3b-CHAPk ranged from 0.5 to 125 μg/mL. Moreover, 100 μL of prepared bacterial solutions were added to the different wells containing 100 μL of the different concentrations of either CHAPk or CHAPk-SH3blys. Plates were incubated for 16 h at 37 °C, after which individual wells were visually inspected for growth. Moreover, 100 μL aliquots were removed from clear wells, spread onto MSA, and then incubated for 16 h at 37 °C to confirm the absence of viable bacterial cells. This experiment was performed using three biological replicates.

### 4.6. Staphylococcal Biofilm Assays

The microtiter plate assay described by O’Toole [44] was used to analyze biofilm formation of *S. aureus* DPC5246, with minor modifications. Briefly, log-phase and 7-day-old cultures of *S. aureus* DPC5246 (prepared as described in Section 4.3) were diluted in BHIg to a starting inoculum of approximately 1 × 10^6^ CFU/mL and 200 μL volumes of the prepared cultures were aliquoted into wells of a sterile flat-bottomed 96-well microtiter plate (Sarstedt, Newton, NC, USA) and incubated at 37 °C for 16 h. After this incubation period, the contents of the wells were removed, and the wells were washed three times with 225 μL of 0.8% NaCl. The plate was inverted and left to dry for 60 min at 50 °C. The biofilms were then stained with 200 μL of 0.1% crystal violet solution (Sigma) for 15 min at room temperature. The stain solution was removed, and the wells were gently washed as before. The plate was left to dry at room temperature, after which 30% acetic acid (Sigma) was added to solubilize the stained biofilm. The plate was refrigerated at 4 °C for 30 min. A total of 100 μL from each well was transferred to a fresh microtiter plate and OD_595_ readings were taken using a plate reader (Spectra MAX 340, Molecular Devices, Molecular Devices, San Jose, CA, USA).

### 4.7. Staphylococcal Biofilm Reduction Using CHAPk and CHAPk-SH3blys

*S. aureus* DPC5246 biofilms were established using log-phase and 7-day-old cultures as starting inoculum as described in Section 4.6. Biofilm containing wells were washed as described in Section 4.6 and then treated with 200 μL of various concentrations of CHAPk and CHAPk-SH3blys (3.91–250 μg/mL) in sterile 25mM Tris pH 7 or with 200 μL of sterile 25 mM Tris pH 7 alone (control), at 37 °C for 4 h. At the end of treatment, all wells were washed again before the plate was inverted and left to dry for 60 min at 50 °C. The biofilms were then stained with 200 μL of 0.1% crystal violet solution for 15 min. The stain solution was removed, and the wells were gently washed as before. The plate was left to dry, after which 30% acetic acid was added to solubilize the stained biofilm. The plate was refrigerated at 4 °C for 30 min. A total of 100 μL from each well was transferred to a fresh microtiter plate. The OD readings were carried out at 595 nm using a plate reader.

### 4.8. Reproducibility and Statistical Analyses

All experiments were performed in triplicate using three biological repeats. All comparisons between data were based on the mean ± standard deviation. Two-way ANOVA and Tukey pairwise comparisons were carried out. Statistical analysis and graphs were generated using GraphPad Prism 9.1.0 (GraphPad Software Inc., San Diego, CA, USA). In all cases, a *p* value < 0.05 was established as a threshold to consider a result as statistically significant. 

## Figures and Tables

**Figure 1 antibiotics-10-00722-f001:**
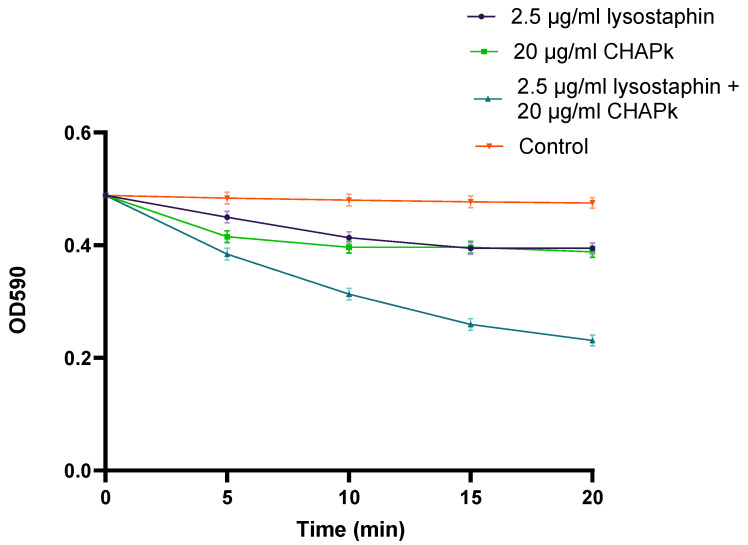
Turbidity reduction assay employing CHAPk (20 μg/mL), lysostaphin (2.5 μg/mL) and a combination of both (20 μg/mL of CHAPk with 2.5 μg/mL of lysostaphin) against stationary phase cultures of *S. aureus* DPC5246. Cells were also treated with 25mM Tris-HCl pH 7 (control). Each OD_590_ measurement is the average of triplicates plus/minus the standard deviation.

**Figure 2 antibiotics-10-00722-f002:**
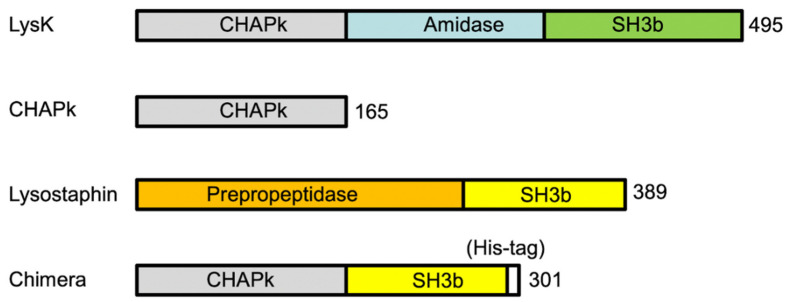
Schematic representation of the enzymatic domains of LysK, Lysostaphin, CHAPk, and CHAPk-SH3bLys chimera. LysK is composed of two different catalytic domains (CHAP and amidase) and a cell-binding domain (SH3b). Lysostaphin is composed of a peptidase catalytic domain and a cell-binding domain (SH3b). CHAPk is composed of the CHAP domain of LysK. CHAPk-SH3bLys is the chimeric protein generated for this study, and it is composed of the CHAP domain from LysK and the SH3b cell-binding domain from lysostaphin.

**Figure 3 antibiotics-10-00722-f003:**
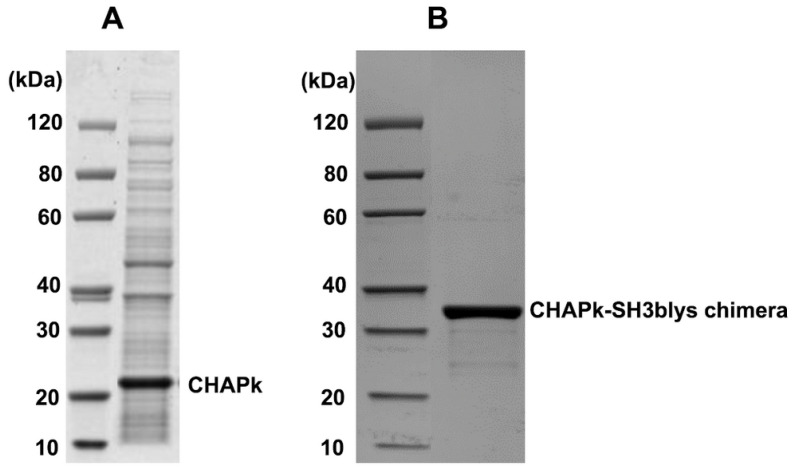
A 12% SDS-PAGE gel with (**A**) CHAPk enzyme preparation. A band of approximately 20 KDa was observed, which corresponds to the predicted molecular mass of 18.6 kDa; (**B**) CHAPk-SH3blys enzyme preparation. A band of approximately 35 kDa was observed, which corresponds to the predicted molecular mass of 33.6 kDa.

**Figure 4 antibiotics-10-00722-f004:**
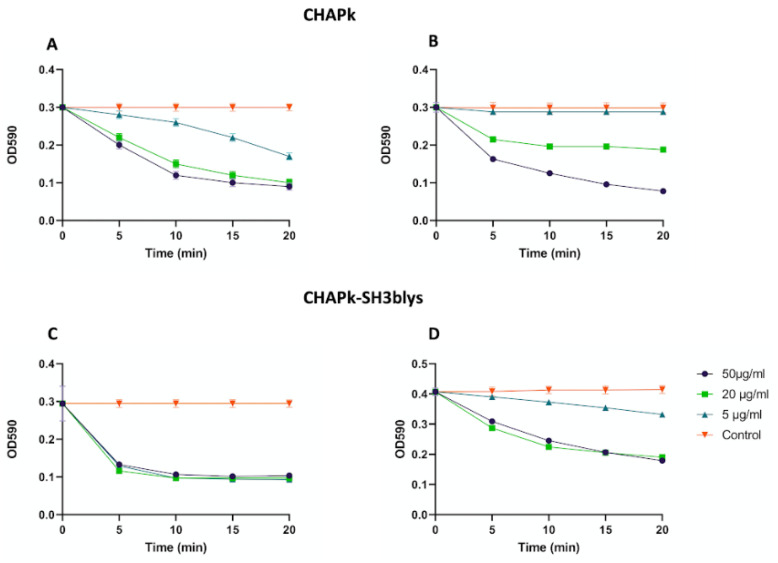
Turbidity reduction assay employing (**A**) CHAPk final concentrations of 5, 20, and 50 μg/mL against log-phase cultures of *S. aureus* DPC5246; (**B**)CHAPk final concentrations of 50, 20, and 5 against 7-day-old cultures of *S. aureus* DPC5246; (**C**) CHAPk-SH3blys final concentrations of 50, 20, and 5 against log-phase cultures of *S. aureus* DPC5246; (**D**) CHAPk-SH3blys final concentrations of 50, 20, and 5 against 7-day-old cultures of *S. aureus* DPC5246. Cells were also treated with 25 mM Tris-HCl pH 7 (control). Each OD_590_ measurement is the average of triplicates. Error bars represent the standard deviation.

**Figure 5 antibiotics-10-00722-f005:**
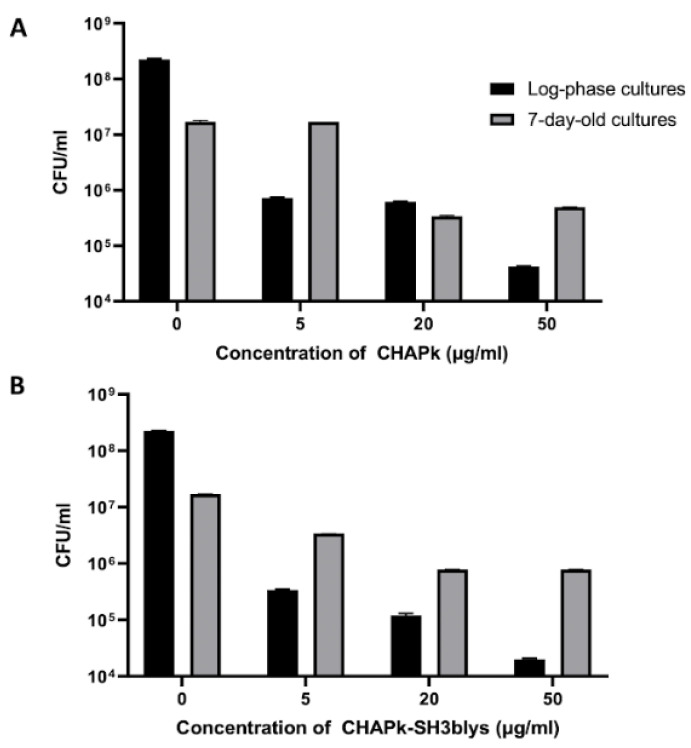
Results for viable plate assay (CFU/mL) after the treatment of log-phase and 7-day-old cultures with 5, 20, and 50 μg/mL of (**A**) CHAPk or (**B**) CHAPk-SH3blys for 20 min at 37 °C. Each CFU/mL value is the average of triplicates plus/minus the standard deviation.

**Figure 6 antibiotics-10-00722-f006:**
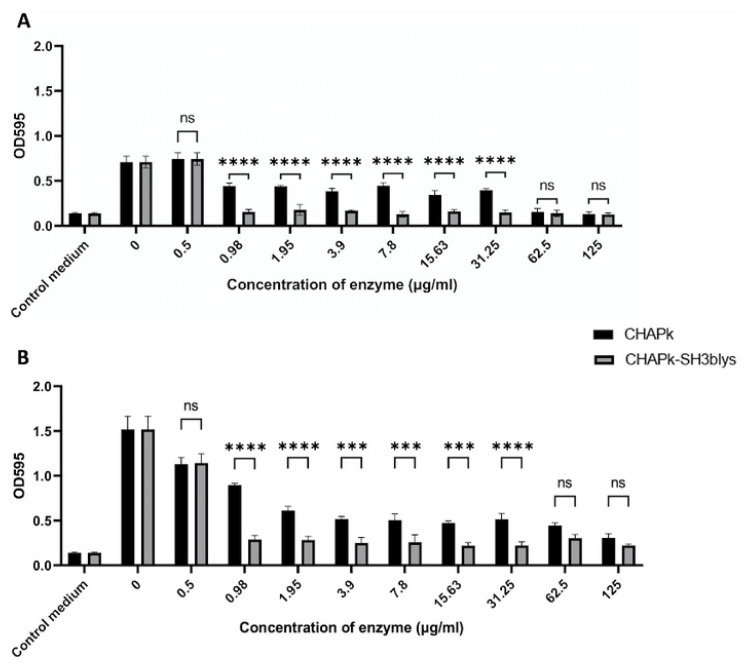
Biofilm prevention assay using concentrations of CHAPk and CHAPk-SH3blys ranging from 0.5 to 125 μg/mL for (**A**) an initial inoculum of log-phase cultures or (**B**) an initial inoculum of 7-day-old cultures. OD_595_ readings are the average of triplicates plus/minus their standard deviation. *p*-values <0.001 are represented by ** and *p*-values <0.0001 are represented by ***. No significant statistical differences are represented by ns.

**Figure 7 antibiotics-10-00722-f007:**
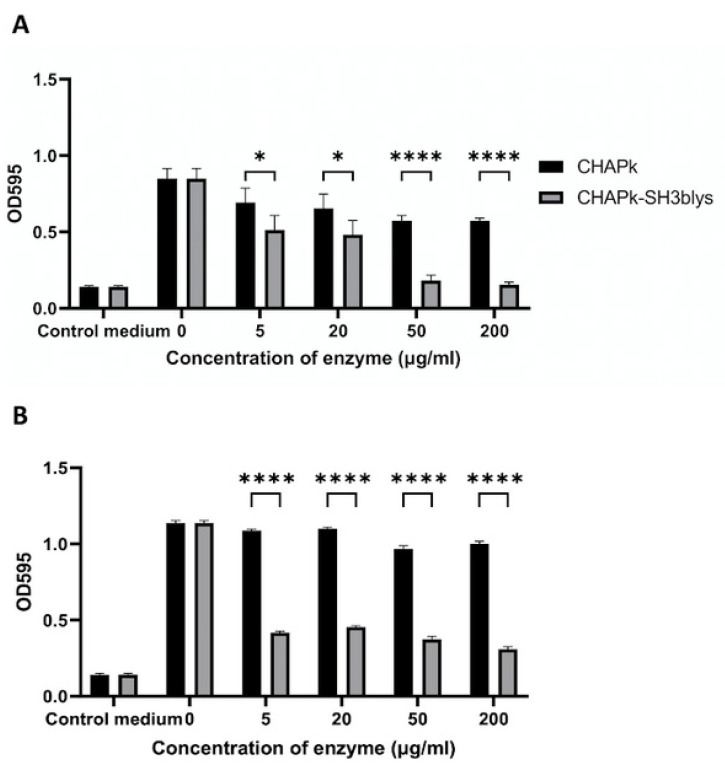
Biofilm disruption assay using concentrations of CHAPk and CHAPk-SH3blys ranging from 5 to 200 μg/mL for (**A**) an initial inoculum of log-phase cultures or (**B**) an initial inoculum of 7-day-old cultures. OD_595_ readings are the average of triplicates plus/minus their standard deviation. *p*-values < 0.0001 are represented by ****. No significant statistical differences are represented by ns. *p*-values for comparing CHAPk-SH3blys and CHAPk concentrations of 5 and 20 μg/mL (for biofilms formed from log-phase cultures) were 0.0115 and 0.0160, respectively (both represented by *).

**Table 1 antibiotics-10-00722-t001:** Minimal inhibitory concentration (MIC) values for CHAPk or CHAPk-SH3blys against log-phase and 7-day-old cultures of *S. aureus* DPC5246.

	MIC (μg/mL)	
Enzyme	Log-Phase Cultures	7-Day-Old Cultures
CHAPk	31.25	125
CHAPk-SH3blys	7.8	7.8

## Data Availability

All the data are available.

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
