# Peer review of "Engineering of the CHAPk Staphylococcal Phage Endolysin to Enhance Antibacterial Activity against Stationary-Phase Cells"

_antibiotics, 2021, doi:10.3390/antibiotics10060722_

Round 1

Reviewer 1 Report

The manuscript: "Engineering of the CHAPk staphylococcal phage endolysin to enhance antibacterial activity against stationary-phase cells." discusses preparation and application of chimeric enzyme designated CHAPk-SH3blys combining a truncated derivative of staphylococcal phage K endolysin CHAPk with the cell-binding domain of the bacteriocin lysostaphin. The authors show that this recombinant enzyme can prevent and disrupt MRSA biofilms more efficiently than CHAPk alone. This topic is of high importance since the antibiotic resistance is becoming major problem globally and alternative ways to combat multiresistant bacteria are needed. All in all the manuscript is well written and the experiments well conducted. However, I have some suggestions:

You could mention that it was His-tag affinity purification (I am assuming based on the plasmid that you used). It is not specified in methods section. Also, which affinity columns were used etc.

Figure1: Are standard deviations missing from the figure itself?

Figure 2 b is quite poor quality (or Figure 3). Could the gel be run again?

There is no figure 3 and instead two figure 2.

Figure 4: Maybe the name of used enzyme could be mentioned in the picture and also the phase of the culture. It might be more clear for the reader to see the results.

Figure 5: Please, mention about the error bars and how many repeats. The results could be demonstrated also directly as fold differences compared to 0 in the diagram. In my opinion this would make the figure more readable and easier to compare the differences.

Some spelling mistakes:

row 166: MSA is mentioned here the first time. Please add the explanation

row 185: triplicate --> triplicate.

row 190: remove one extra..

row 214: remove extra all

row 209: There is no figure 8?

row 245: Is there two spaces after Also,  ?

row 249: required.Phage --> required. Phage

row 278: No space after [37].

Author Response

I have attached one document with comprehensive replies (in red) to both referees.

Reviewer 2 Report

The work describes the antibacterial activity of CHAPk and lysostaphin individually and in chimeric protein, CHAPk-SH3blys, that they product. In particular the results highlight CHAPk-SH3blys exhibited a 16-fold lower MIC than CHAPk. Moreover, low concentrations of CHAPk-SH3blys, 1 μg/ml and 5 μg/ml respectively, are enough to prevent and disruption biofilm, whereas for CHAPk, 125 μg/ml was needed.

Although the manuscript is quite organized, some results and discussions are missing, then I suggest revisions:

Major revision:

  • In the results the Author compare the data obtained using CHAPk-SH3blys with CHAPk, however the comparison should be carried out also with a mixer of CHAPk and lysostaphin in the same molar concentration of the chimeric protein.
  • In the section “discussion”: The lines 244-264 should be considered a part of introduction; The lines 272-283 are a summary of the results. The real discussion is only in lines 265- 271

I suggest to the Author to increase the discussion by comparison of each obtained data with those from other authors (even just for those in log-phase).

  • The Authors in lines 268-271 proved to explain the originality of their approach. However, only 3 lines are not enough, I suggest the Author to implement the motivations that make their work original.

Minor revision:

  • In line 111, check the denomination “Figure 2”. It should be change in “Figure 3”.
  • The figures 4, 6 and 7 should be have the same scale to better visualize the results.
  • Figure 2 and related discussion (lines 91-93) should be moved in section Material and method.
  • In figure 6 and 7, the control medium could be removed because the real control is the biofilm formation at 0 µg/mL concentration.
  • The supplementary materials are missing.
  • In line 230, check the denomination “figure S78”.
  • All the figures should be improved in the resolution.

Author Response

I have replied to both refereed in the same (attached) document
